# Tourists perceptions of Caribbean islands facing environmental threats before the COVID-19 health crisis: Holbox Island and Archipelago of Bocas del Toro

Nadia T. Rubio-Cisneros[1,2,3‡]*, Jorge L. Montero-Muñoz[4], Igor I. Rubio-Cisneros[3,5], Sara Morales-Ojeda[3,4], Mónica Pech[4,6], Gabriel Ruiz-Ayma[1,3], Marisol Rueda-Flores[3,7], Rachel Baker[3,8], Armando Jiménez[1,3], Karen Fuentes[9], José I. González-Rojas[1‡]*

1 Laboratorio de Biología de la Conservación y Desarrollo Sustentable de la Facultad de Ciencias Biológicas, Universidad Autónoma de Nuevo León, Monterrey, Mexico, 2 Fellow for Coastal Ecosystem Services, Center for Marine Biodiversity and Conservation, Scripps Institution of Oceanography, University of California San Diego, La Jolla, California, United States of America, 3 Mar Sustentable Ciencia y Conservación, A.C., Mexico, 4 Centro de Investigación y de Estudios Avanzados del Instituto Politécnico Nacional (Cinvestav), Unidad Mérida, Mexico, 5 Secretaría de Extensión y Cultura, Colegio Civil Centro Cultural Universitario s/n, Universidad Autónoma de Nuevo León, Monterrey, Nuevo León, México, 6 School of Earth and Sustainability, Northern Arizona University, Flagstaff, Arizona, United States of America, 7 Healthy Reefs for Healthy People, United States of America, 8 University of North Carolina (NC) Wilmington, Wilmington, North Carolina, United States of America, 9 Manta México Caribe, A.C., Mexico

‡ NTRC and JIGR are joint senior authors on this work.
* rubio.nadiat@gmail.com, nadia@marsustentable.org (NTRC); jose.gonzalezrjs@uanl.edu.mx, josgonza@gmail.com (JIGR)

## Abstract

Knowledge gaps exist in the socio-ecological systems of small touristic islands in Latin America. Understanding tourists' perceptions of their environmental knowledge can help plan actions to prevent natural capital loss necessary for local economies. Tourists' perceptions of a touristic hotspot, Holbox Island, were documented. Surveys demonstrated that tourists are aware of their environmental impacts and are interested in minimizing these. Results were compared with results on Bocas del Toro, Panama. Tourists' perceptions had similarities among sites driven by similarities in tourists' populations with a common geographic origin. Tourists lack site-specific knowledge to steer them towards environmentally conscious decisions in both regions. Findings suggest the need to promote local actions to gain tourists' understanding about their destination and support education programs on island conservation. Addressing these needs can help build resilience to overcome the adverse socio-environmental effects of tourism, environmental disasters, and health crises as COVID-19 on small islands.

## Introduction

Here we report on Holbox Island (hereafter Holbox) tourists' perceptions of their environmental knowledge documented via semi-structured surveys and compare our results with our

**Data Availability Statement:** The data is publicly available at http://eprints.uanl.mx/22696/.

**Funding:** We want to thank The Rufford Foundation, UK. for providing the funding for this work as a research grant for Dr. Nadia T. Rubio-Cisneros." The funders had no role in study design, data collection and analysis, decision to publish, or preparation of the manuscript.

**Competing interests:** The authors have declared that no competing interests exist.

published work on the archipelago of Bocas del Toro Panama (hereafter ABT). Both sites share socioenvironmental threats from tourism overcrowding and natural resource exploitation (e.g., overfishing and land transformation [1–4] (Fig 1).

The research initiated before the Coronavirus Disease (COVID-19) times. Since 2017, our main research drive has been to generate data from tourists' perceptions that could help establish the importance of being informed in the destination visited when mobility was not an issue [5]. This information is needed to propitiate positive interactions between tourists, nature, landscapes, and the local communities in contemporary hotspots for sun and sand tourism facing overcrowding [6]. Nowadays, COVID-19 established in society the breakthrough of the ¨lockdown¨ and the ¨New Normal¨, which have impacted human traveling and have made it a must to be an informed traveler for life survival reasons [7–9]. But the above can also become a turning point in taking advantage of the COVID-19 crisis to become an informed tourist towards the ecological and social challenges that popular tourist destinations face. It could also be an opportunity for tourists to start choosing another kind of tourism that is more friendly to the environment and supports locals, such as community-based tourism [10, 11]. The above can help achieve the environmental goals needed to establish sustainable tourism practices [12].

Sun and sand tourism is widespread in the tropics [13]. Consequently, many small islands face burgeoning tourism economies, triggering a social transformation of fishing communities into popularly global touristic sites [2, 6, 14]. Tourism is growing especially on small islands such as Holbox with attractive landscapes and outdoor activities for international tourists such as snorkeling tours with whale sharks [15, 16]. For example, in November 2021, Idelfonso Cetina Alcocer, an authority of Holbox, said that many islanders left fishing given the fall in catches. Most fishers are working on tourism activities, which provides more profits [17].

An increase in tourism implies modifying landscapes and natural resource exploitation [18]. A rise in these activities becomes a problem for islands that face poor natural resource management, unplanned urban development, and mild environmental enforcement by authorities [3–5, 14, 19]. Consequences of the unregulated anthropogenic transformation include the loss of ecological functions necessary to accomplish ecosystem services on which

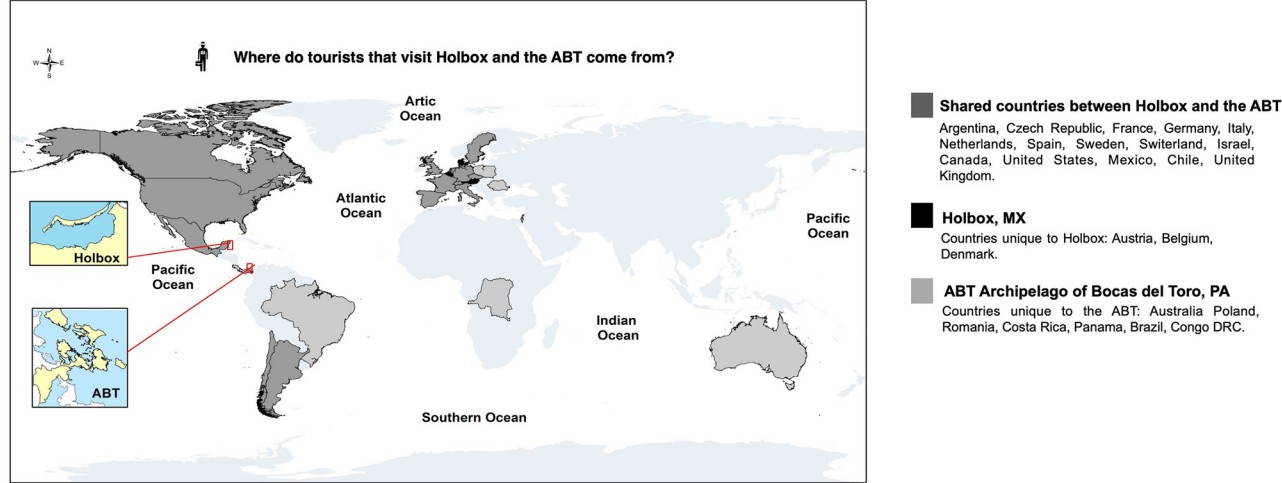

**Fig 1. Where do tourists that visit Holbox and the ABT come from?.** A global map that shows the country of origin of tourists that visit Holbox and the ABT. Fig 1 the original idea is by NTRC and built with personal data and data results from tourists' surveys collected in this research. SM created and designed the map.

the economies of touristic islands rely (*e.g.*, fisheries, biodiversity, coastal protection, carbon sequestration, and water filtration) [9, 20–25].

Socioecological data on tropical islands related to human-nature interactions such as tourists and nature is scant for many regions including Latin America, but necessary for a sustainable management of natural resources [4, 26, 27]. In the case of tourism on small islands, besides being a source of economic revenues for local and national economies, tourism is also a source of broadly disseminating sociocultural values between locals and visitors worldwide [28]. Furthermore, the impact tourists make on their destinations is also an open subject of study in the tourism scholar world [29–32].

## Tourism in Quintana Roo and the current health crisis

The United Nations policy brief on COVID-19 health crisis effects on tourism states that COVID-19 negatively impacted this sector since the tourism network involving economies, livelihoods, and public services lost functionality worldwide [28, 33]. Mexico is among the top ten tourist destinations; the country ranked 7th globally in 2018 and 2019 with 41.3 and 45 million international tourist arrivals, respectively [34]. In 2019 Mexico's tourism industry contributed to 8.7% of the country's GDP and generated 2.3 million direct jobs, each of which produces about 4 to 5 indirect jobs [35]. Mexico's global ranking in tourism is of high importance to destinations like Cancun and The Riviera Maya in the state of Quintana Roo, which in the past 40 years have concentrated massive sun and sand tourism [18]. In 2018 Quintana Roo had 32.4% ($6,902.50 million dollars) of the total tourism revenues for Mexico ($21,322.80 million dollars); this year, the state harbored 14,282,738 tourists and 7,058,561 cruise tourists [36].

Before the COVID-19, Quintana Roo´s coastal territory in Latin America and the Caribbean, was already facing adverse socioenvironmental events related to mass tourism such as extensive land refill and urban development that modified nearshore environments (e.g., mangroves, coastal lagoons, and coral reefs) that are key for the delivery of water, biodiversity, coastal protection and carbon sequestration ecosystem services [37–46].

However, Quintana Roo's large-scale unsustainable tourism activities worked, given its economic profitability for 2019; $15440.41 million dollars were obtained from tourism revenues [47]. The contamination by COVID-19 caused a socioeconomic collapse in the state's main touristic hotspots (e.g., Cancun, Riviera Maya, Isla Holbox, and Cozumel) [33, 35, 48–53]. In June 2020, Quintana Roo, as part of Latin America and the Caribbean, was in the epicenter of COVID-19, where more than 4 million deaths were documented. At that time, that was more than 27% of cases in the world [7] (Fig 2). Furthermore, Mexico reached the Americas' top countries with the highest case fatality ratio (of 9.1 of 10) [9, 54].

For Quintana Roo COVID-19 health crisis also ignited the loss of over 100,000 formal jobs in July [35, 55, 56] (Fig 2). By November 2020, hotel occupancy's aggregated values decreased by 34.9% in Cancun and by 52.3% in the Riviera Maya, compared to values in year 2019 [57]. On average, hotel occupancy in the region is 80% [58].

In June, Mexico started a traffic light system for the resumption of socioeconomic activities. Quintana Roo started a regional traffic light system for the northern (Lázaro Cárdenas (where Holbox is located), Tulum, Solidaridad, Cozumel, Puerto Morelos, Benito Juarez, Isla Mujeres); and southern (Felipe Carrillo Puerto, José María Morelos, Bacalar, Othon P. Blanco) municipalities (Fig 2). As of July, Quintana Roo reopened for tourism, and by October 2020, hotel occupancy began to recover [59]. However, nationally, challenges still exist for the tourism sector due to the lack of federal support in the country's economic reactivation plans [35, 60]. Challenges are faced regionally as well, specifically in coastal towns of Quintana Roo while

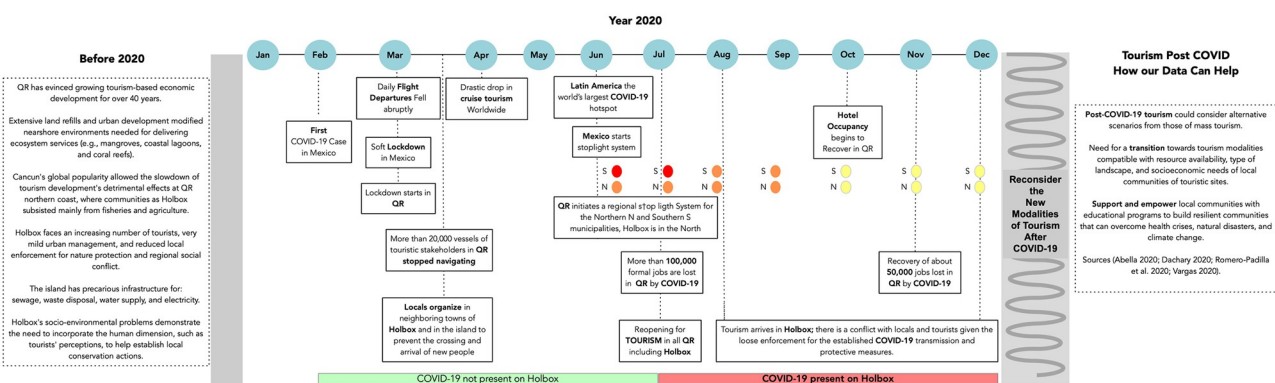

**Fig 2. Social and economic events related to COVID-19 afflicted locals and tourists in Quintana Roo (QR) and Holbox.** Mexico's health authorities implemented a national stoplight system in June 2020 to alert residents to the epidemiological risks and provide guidance on restrictions on certain activities. The traffic light system includes four colors: green, yellow, orange, and red. Green indicates a low risk of contagion, and all activities, including school, are allowed. Red indicates a maximum risk of contagion, and only essential economic activities will be allowed. Quintana Roo started a regional traffic light system for the northern (Lazaro Cardenas (where Holbox is located), Tulum, Solidaridad, Cozumel, Puerto Morelos, Benito Juarez, Isla Mujeres); and southern (Felipe Carrillo Puerto, Jose Maria Morelos, Bacalar, Othon P. Blanco) municipalities. The original idea and creation of the figure are by NTRC. Sources for Fig 1 [35, 48–53, 56, 59, 61, 86, 121–123].

locals try to regain tourism activities. These include noncompliance with the health measures associated with COVID-19 [61], social conflict related to the long-term lack of income within locals, and sociocultural and cultural heritage losses occurring in coastal towns by the deaths of elderly inhabitants. For example, our fieldwork on Holbox in 2021 documented that COVID-19 has severely afflicted the tiny population of elder fishers inhabiting Holbox. Their local knowledge was a source for historical reconnaissance of events related to fishing and changes in the island´s landscape by anthropogenic actions.

Our work on tourism perspectives of the environment can help natural resource and urban development managers understand the human dimension of nature enjoyment from a pre COVID-19 point of view and pave ways to highlight changes accordingly to the ¨New Normal¨ where being informed of our surroundings is now a must. This paper aims to (1) improve the knowledge of a regional problem of scarce baseline documentation of tourists on islands becoming hotspots for global tourism; (2) demonstrate how the information of tourists surveyed may guide the development of regional conservation policies and environmental education for preserving coastal ecosystem services and (3) lead ways for educating people to become an ¨informed tourist¨ to improve human-nature and social interactions in the visited destinations during or after a health crises.

## Methods

### Study sites

Holbox is located off the northern coast of the Yucatan Peninsula in the state of Quintana Roo, Mexico. It is a 42 km long and 2 km wide barrier island that is intermittently connected to mainland. Holbox's population is ~1,143 people, but throughout the high season, from May to September, a floating population of over 10,000 people demands services [62] (Figs 3 and 4). However, the island has a precarious infrastructure for sewage and trash disposal, freshwater supply, and electricity. Holbox's rapid and unorganized urban and tourist development put the Island's environmental health at risk [4, 63–65]. The island harbors critical habitats (e.g.

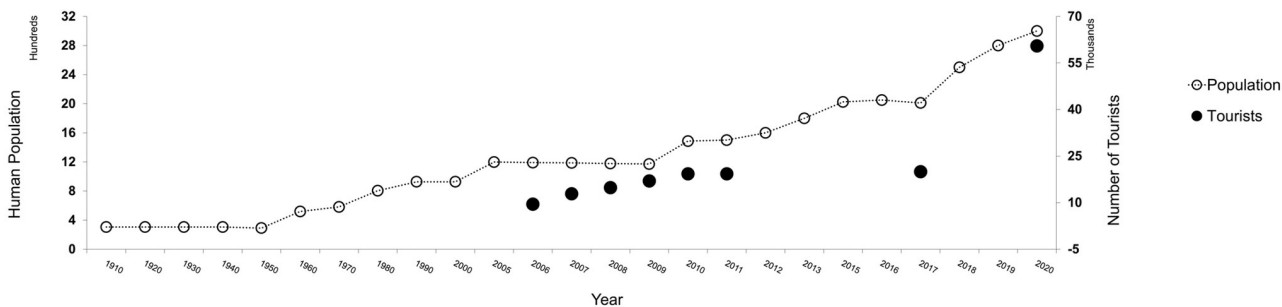

**Fig 3. Holbox human population and available data for tourists' numbers.**

water springs, mangroves, coral reefs, and seagrasses) and species of flora and fauna for conservation; for example, red mangrove (*Rhizophora mangle*), black mangrove (*Avicennia germinans*), loggerhead sea turtle (*Caretta caretta*), hawksbill sea turtle (*Eretmochelys imbricata*), American Crocodile (*Crocodylus acutus*), American flamingo (*Phoenicopterus ruber*), roseate spoonbill (*Ajaia ajaja*), Northern Tamandua anteater oso (*Tamandua mexicana*), jaguar (*Panthera onca*), puma (*Puma concolor*), and manatee (*Trichechus manatus*) [66].

The protection of Holbox's nature exists through various Mexican laws and the Yum Balam Natural Protected Area (NPA). However, the area's management plan was published in 2018, twenty-four years after the decree of the NPA [67].

Holbox's results are compared with those in the archipelago of Bocas del Toro Panama (hereafter ABT), a place renowned for its biodiversity [68] (Fig 1). A National marine park created in 1988 protects fauna and landscapes. However, the once abundant lobsters, conchs, sharks, and many commercial fish suffer overexploitation [1, 69, 70]. Anthropogenic actions increased in severity in the ABT as "sun and sand tourism" expanded in the Caribbean and it became a Zone of Tourism Development of National Interest for Panama [13, 71, 72]. The

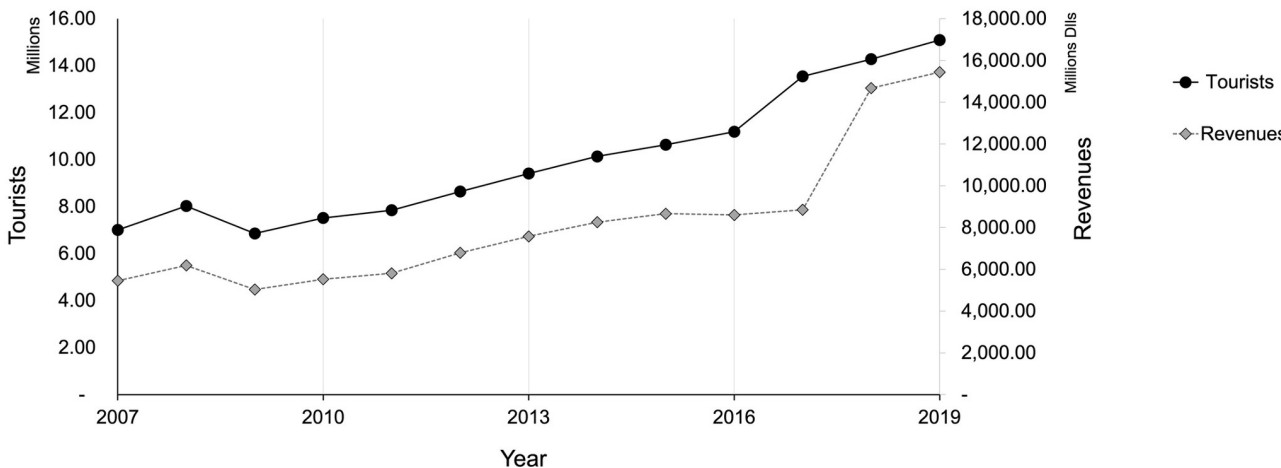

**Fig 4. Tourists that visited the state of Quintana Roo and revenues generated by these visitors.** Sources for the figure [36, 47, 57, 62].

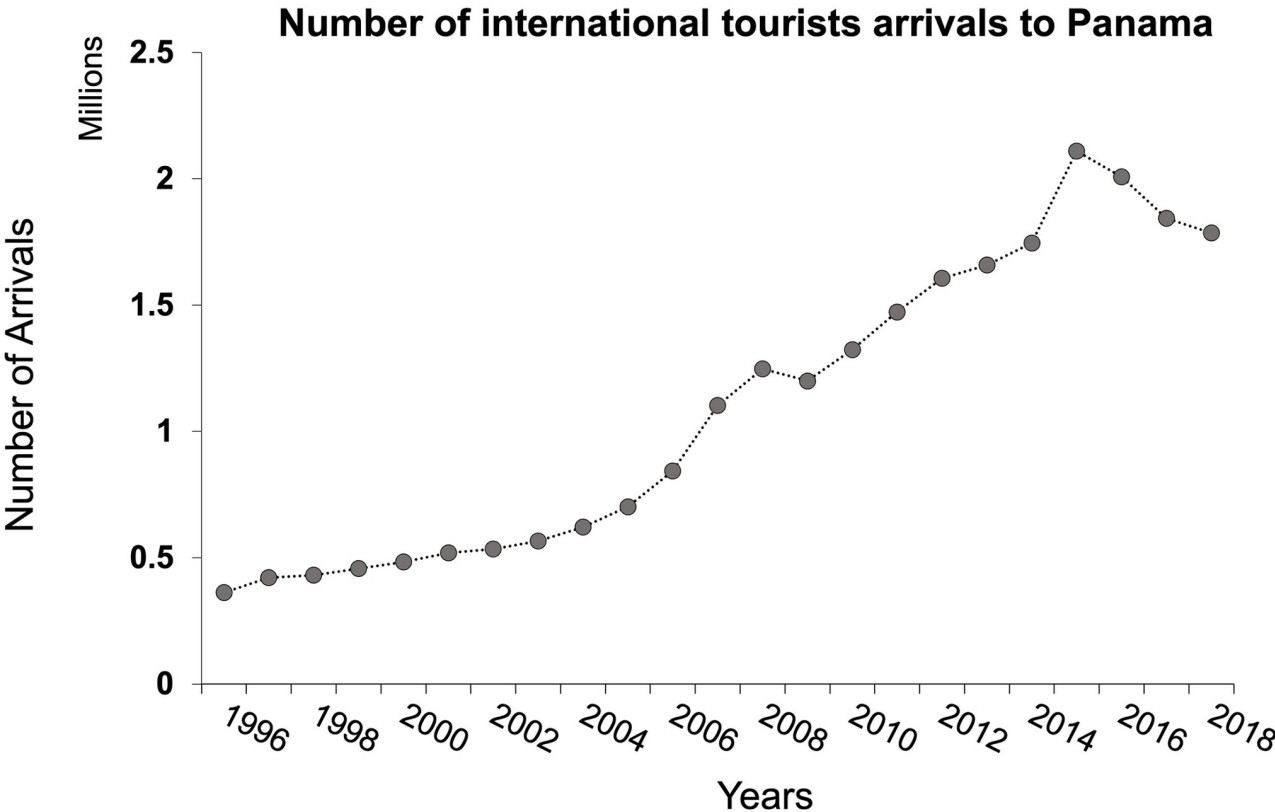

**Fig 5. Tourists' arrivals to Panama.** Specific annual numbers of tourist arrivals to the ABT could not be obtained. Source for the figure The World Bank [124].

increase of tourists visiting the ABT as well as new residents to the area are drawn in by the economic opportunities of tourism, leading to accelerating growth rates in human population (Fig 5). This growing population has consumption needs that threaten the availability of ABT's natural resources, which provide vital ecosystem services for the local economy [73, 74].

## Field work

**Tourist surveys.** Open interviews from key community members were collected (e.g., native residents, community leaders, conservationists, and elder fishers), to acquaint the authors with socioenvironmental issues and people´s perspectives of the natural environment in the region. We used the survey approach of Dorsett & Rubio (2019) [5] for documenting tourists' perceptions of the environment in ABT. Tourists were surveyed with a structured questionnaire, by [5]. This focused on the interviewee's knowledge and opinions of the environmental impacts of tourism, the decisions the interviewee has made with regard to their environmental impact during their time on Holbox, and the interviewee's engagement with seafood consumption during their visit (details of the survey are published in [5]). The survey method utilized was convenience sampling of as many individuals as possible that identified themselves as tourists [75, 76]. The potential bias of convenience sampling was mitigated by a large sample size (n = 230). The human subjects that participated in this study were tourists visiting Holbox Island. All the tourists from which we obtained a survey gave the researchers

oral consent to participate. For interviewing tourists, we followed published guidelines and methodology from our study on Bocas del Toro, Panama [5]. The survey methodology of this study is approved by the Ethical and Research Committee of Health Services from the State of Colima, Mexico (Servicios de Salud del Estado de Colima Instituto Estatal).

### Data analysis

Data from the tourists´ surveys were analyzed by summing up the number of responses for each answer category of each survey question. All information was plotted into bar charts (Figs 6 and 7). To further explore the differences between the responses of tourists in Holbox and ABT, we used a database of tourists' responses as descriptors and geographic sites as units of the analysis. With the database, a triangular matrix of dissimilarity was built using the Percentage Difference (*alias* Bray-Curtis index). The dissimilarity matrix was ordered using non-metric multidimensional scaling (NMDS) [77, 78] to represent the dissimilarity between objects (geographic sites) in an ordination plot (Fig 8A).

The Analysis of similarities (ANOSIM) [79] was used to test statistically whether there is a significant difference between the responses of tourists in Holbox and ABT. The test uses distance measures converted to ranks, and the test statistic R ranges from 0 to 1, with a larger, positive value for R signifying dissimilarity between groups. Lastly, the comments that went along with question 5 of the tourists' survey were arranged in categories using a pie chart (Fig 9). The pie chart results were integrated and discussed within a framework of the future delivery of coastal ecosystem services. All the previous analyzes were performed using the PREMIR v7 computer program [80].

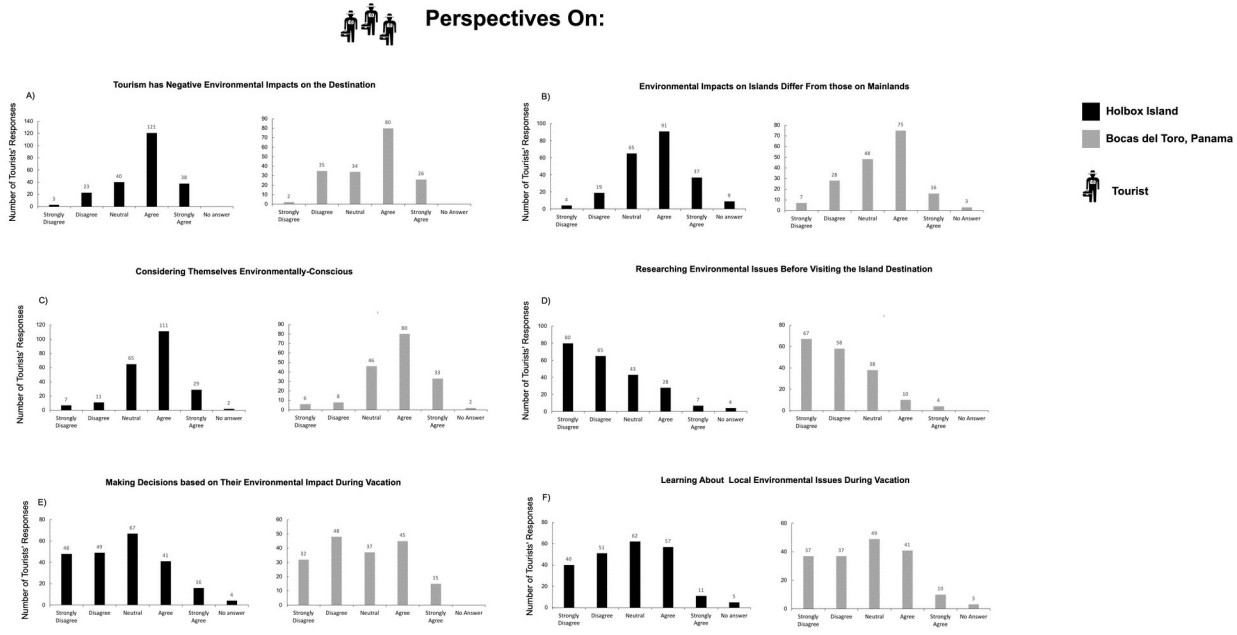

**Fig 6. Bar charts depict tourists' responses to the field survey.** (A) tourism negative environmental impacts; (B) tourism environmental impacts between islands and mainland; (C) tourists who consider themselves environmentally-conscious; (D) tourists who researched on their lodging, food choice and leisure activities before visiting the archipelago; (E) tourists who made decisions based on their environmental impact during vacation; (F) tourists who learned about local environmental issues during vacation.

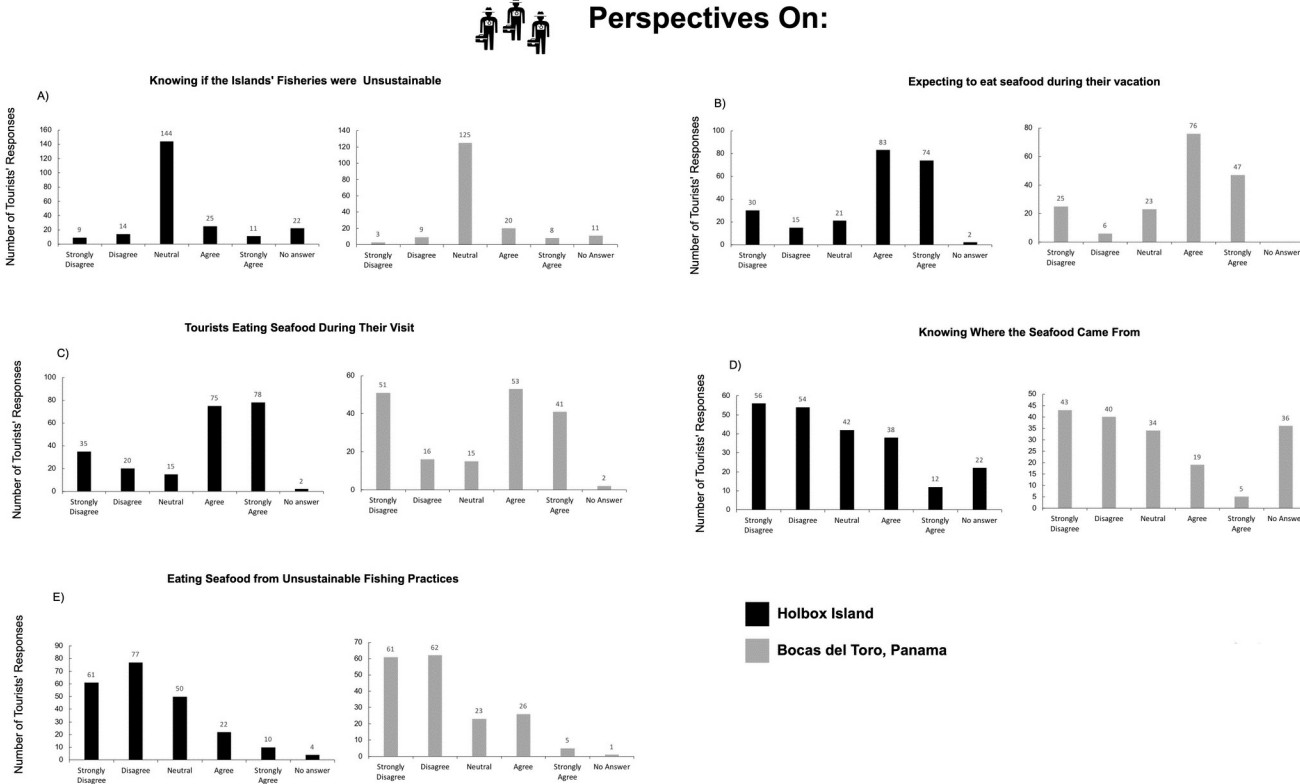

**Fig 7. Bar charts depict tourists' responses to the field survey.** (A) Holbox fisheries; (B) tourists' expectations of consuming and eating seafood (C); (D) tourists who knew where the seafood came from; (E) tourists who would eat unsustainable seafood.

## Results

### Tourist perceptions of their environmental impact and knowledge on Holbox

The results show that in general, tourists are very aware that tourism has negative environmental impacts on the destination (Fig 6A). Even further, over half of tourists "Agree (n = 91)" or "Strongly Agree (n = 37)" that as an island destination, the environmental impacts of tourism on islands differ from those on mainland destinations (Fig 6B). The majority of tourists (n = 111) would consider themselves to be environmentally-conscious (Fig 6C), but very few agreed (n = 28) that they researched environmental issues of Holbox before their vacation (Fig 6D). Moreover, less than one-third of respondents agreed (n = 41) or strongly agreed (n = 16) that they considered environmental impacts when they made decisions about their lodging, food choice, and leisure activities during their stay on Holbox (Fig 6E). From the above, fifty tourists filled out the survey comment section that asked them to list how they did so. The most frequent comments were "eating local foods and at local restaurants"; "searched for sustainable lodging"; "minimizing waste generation and proper waste disposal"; and "avoiding disturbance of nature". The importance of using friendly environmental sunscreen and water conservation issues was only mentioned twice.

Less than one-third of respondents agreed or strongly agreed (n = 68) that they had learned about some of the local environmental problems so far during their stay in ABT (Fig 6F). When prompted with the follow-up question, "If so, how/from whom did you learn this?", 59

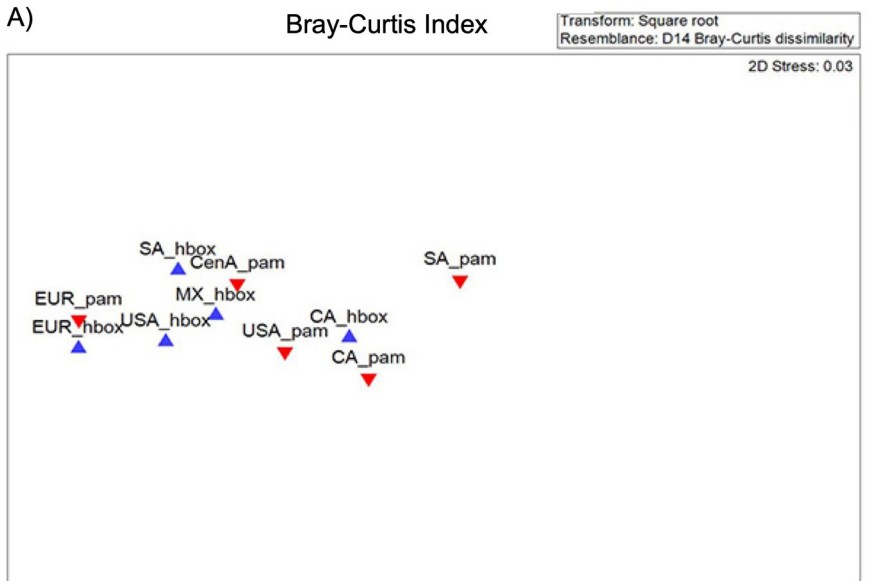

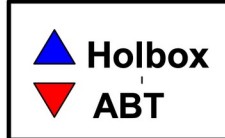

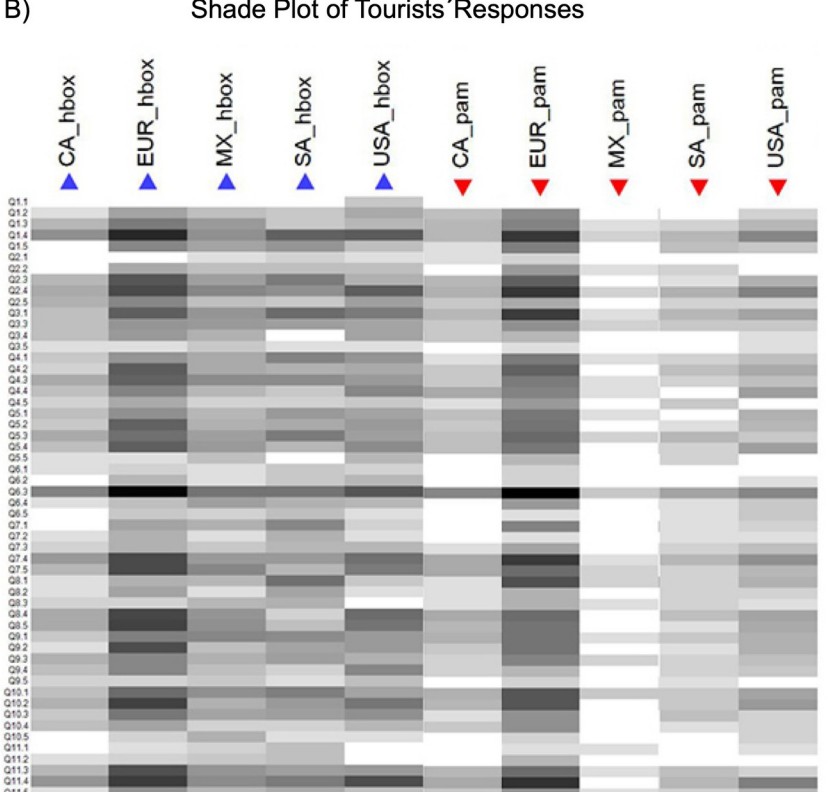

**Fig 8.** (A) Heatmap of tourists" responses between Holbox and the ABT. (B) Bray-Curtis non-parametric analysis. A presence-absence matrix of tourists' responses as descriptors and geographic sites as units of the analysis was used.

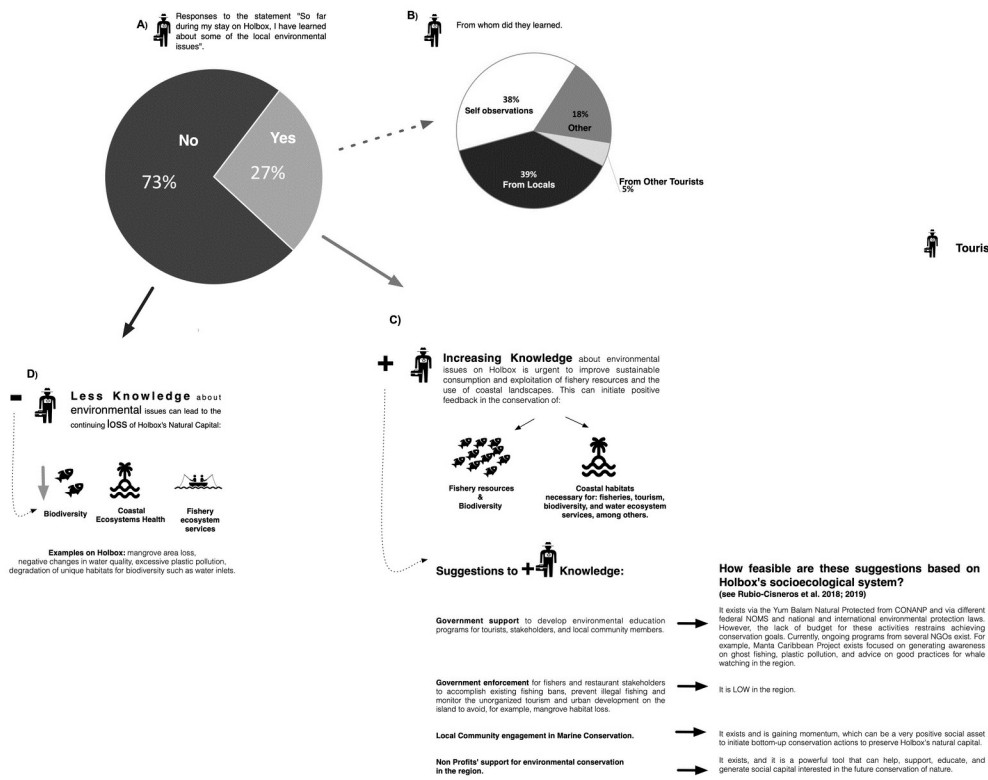

**Fig 9. Tourists´ knowledge about environmental issues on Holbox.** (A) Chart of tourist responses when prompted the question from whom they learned about local environmental issues on Holbox. (B) Categorization of comments that tourists listed when prompted with the follow-up question: If so, how/from whom did you learn this? In response to the statement, "So far during my stay on Holbox, I have learned about some of the local environmental issues" (n = 225). (C) Suggestions to increase tourists' environmental knowledge on Holbox. (D) Consequences of absent tourists' knowledge of environmental issues on Holbox.

people responded and 165 did not. Of those who responded, 39% of respondents commented that they had learned from locals (usually either boat drivers or tour guides), and 38% cited their observations or fellow tourists (Fig 9). This result is reinforced by most respondents who either answered "Neutral (n = 144)" to or left unanswered when asked if they knew if fisheries on Holbox were sustainable (Fig 7A).

Our results also demonstrate 70 percent of respondents agreed or strongly agreed (n = 157; Fig 7B) that they expected to eat seafood during their vacation, and 68% reported to had eaten seafood (n = 153; Fig 7C). Furthermore, only 14% of respondents (n = 50) agreed or strongly agreed that if they have eaten seafood during their time on Holbox they knew where this seafood came from (Fig 7D). However, when tourists were asked if they would still eat a serving of seafood if they knew it came from an unsustainable source only 14% of tourists (n = 32) agreed or strongly agreed that they would still eat seafood (Fig 7E).

## Tourists' perceptions of the environment across geographic regions

Tourists' perceptions of the environment had had low dissimilarities between response patterns when comparing responses from Holbox and from ABT (details on the ABT responses are published in [5]). This last one was our earlier study site. We can see this in Fig 8A in NMDS. The figure shows that tourists from the same initial destination had similar responses

regarding their environmental perspectives for both geographic regions sampled. As well, the pattern of responses had more similarities within the Holbox population than within the Panama population. Each question also analyzed tourists' perceptions responses by region. Results are shown in a shade plot (Fig 8B), a qualitative figure with different intensities of gray. Fig 8B also shows similar color patterns for the different questions among the tourists that have a similar initial destination.

## Discussion

### Tourists and the environment

Tourists' awareness of their negative environmental impacts (Fig 6A) and consideration of being environmentally conscious are social traits that can help reduce adverse anthropogenic effects on Holbox (Fig 6C). Limitations for these traits lie in tourists' few site-specific bits of knowledge on Holbox's environmental issues and their low consideration of their environmental impact when making traveling decisions (Fig 6D and 6E). Additionally, the tourists' short visit limits their understanding of the problem for each destination [81].

Results also display a need to strengthen the communication among "locals-tourists and officials" to pass site-specific information on Holbox's environmental issues (Fig 9). This can give tourists options to minimize their ecological impact.

A problem here is that Holbox's socio-cultural values changed in recent decades. Social organization can influence communication between "locals-tourists." The Holbox region faced increasing coastal migration primarily by the availability of new fishing grounds for non-locals. Additionally, newcomers arrived seeking the economic opportunities of a burgeoning tourism economy, and the inharmonious sellouts of communal land to private investors became widespread. The combination of these factors has fostered social turmoil, which triggered negative changes in human relationships and social-cultural values [see 4]. For example, Holbox's socio-cultural values related to traditional environmental knowledge and self-extractive regulations that once existed are now fading. These social traits are necessary for areas as Holbox, where locals face a continuum of visitors that use and enjoy nature (e.g., on the high season ~ 7000 to 10,000 tourists can daily visit this island daily, but no official statistics are available for recent years).

Nowadays, many islanders and former fishers work on tourism-related jobs, and fishing is broadly considered unprofitable [4, 17]. Former fishers have solid bases on local and traditional ecological knowledge related to the environment [4, 82]. Promoting the sharing of this information between islanders can help enhance islanders' compliance with nature conservation and encourage dialogue between "locals-tourists" (Fig 9). The above thus can also help preserve the continuity of Holbox's socio-cultural values and traditional knowledge between islanders, especially those working on tourism jobs. The attributes of local ecological knowledge combined with scientific research for local or regional conservation need to be considered by resource managers and authorities, for example, for the case of developing sustainable tourism activities for marine conservation [e.g., 83–85].

Positive socio-cultural changes have evolved on Holbox, given increasing environmental threats. Islanders organized conservation groups for sea turtles (Alma Verde), manatees (Grupo Manaholchi), and socioenvironmental issues (Comité Ciudadano por la Paz y Seguridad de Holbox). With the arrival of COVID-19, Holbox's authorities and many citizens organized to prevent visitors and tourists' entry to the island. There was a generalized fear among inhabitants, given the lack of infrastructure and medical personnel on the island. Until then, no cases of the virus existed on the island [86] (Fig 2). The cases of COVID-19 increased on the island in August with the reopening of tourism activities in July [87].

Holbox´s social capital is a resource that authorities and NGO's could use for developing bottom-up conservation campaigns focused on promoting "locals-tourists communication". This task is very necessary nowadays when global changes in the economy caused by the pandemic are opening pathways to reconsider ¨the new modalities of tourism post-COVID-19¨. Some studies by Vargas (2020), Dachary (2020), Hakim (2020) and Sheller (2021) mention that post-COVID-19 tourism could consider alternative scenarios from those of mass tourism [11, 88–90]. For example, a transition towards tourism modalities that are compatible with resource availability, type of landscape, and socioeconomic needs of local communities of touristic sites [9, 10]. The positive socio-cultural changes in Holbox mentioned in the earlier paragraph involve attributes such as resilience, awareness, and compliance, can lead a transition for promoting tourism modalities compatible with resource availability and a culture of conscious tourism on the island [91].

Conscious tourism on the island can involve bottom-up conservation campaigns to promote environmental education programs for islanders to better understand the value of natural capital and provide them with educational tools to inform tourists. Informed locals could inform tourists about pollution, water use, sunblock use, sustainable seafood consumption and help enforce social distancing measures under the ¨New Normal¨. Bottom-up conservation campaigns are critical and have helped in other sites that suffer environmental degradation by tourism or natural resource exploitation. For example, in Cabo Pulmo in the Gulf of California Mexico, locals organized to promote marine conservation and education, to prevent continuing overfishing, large-scale tourism development and organized into small ecotourism cooperatives [92, 93]. Given Holbox's environmental degradation and reduced budget (locally and nationally) to enforce biodiversity loss, creating bottom-up conservation campaigns is critical. Currently, conservation campaigns focused on creating awareness for the links between ecological processes and socioeconomic benefits of islands exist (e.g., NOAA plans for the conservation of barrier islands in the Gulf of México) from which lessons and examples of activities can be learned and tailored for Holbox.

For overcoming COVID-19 global changes in the economy where there is a reconsideration of ¨the new modalities of tourism¨; informed and committed islanders will play a crucial role in conveying to "sell trust" [11, 88, 90]. This objective is considered essential in recent tourism literature for reactivating the local economies of popular touristic sites. For Holbox, this is a challenge since tourists demonstrate a lack of compliance with healthy distance measures from tourists and poor enforcement from authorities, which has generated conflicts between islanders and tourists (Personal Communication).

In Quintana Roo, local campaigns already exist. An example is seen in the Mayan communities in the Sian Ka'an Biosphere Reserve where locals have formed an alliance of sustainable ecotourism cooperatives for over a decade. As part of this alliance, these cooperatives provide community tours with support from the Global Environmental Facility Small Grants Programme. This alliance has generated local jobs through time, which has reduced young people's migration and promoted women's participation in the local tourism economy and community participation in nature conservation [94]. The Manta Caribbean Project has created educational materials to increase locals', visitors', and tour operators' knowledge on the ecology and threats of mobulid rays' populations. This campaign also trains tour operators on good practices for tourism activities related to mantas and whale shark watching and swimming tours (The Manta Caribbean Project https://www.mantacaribbeanproject.org/theproject). "The Refill Project" is another campaign to limit single-use plastics pollution on Isla Mujeres, Quintana Roo by The Manta Caribbean Project and No Mas Plastik. This campaign promotes the refill of people's water bottles in local shops (The link shows a video of the campaign https://www.facebook.com/nomasplastik2015/videos/247392116182805/?t=3.)

Another example is the fishing community in San Crisanto Yucatan, where members-initiated mangrove conservation and restoration by collective land management to protect endemic wildlife [95]. Given the above information, a program can be tailored for Holbox, possibly one that aligns with some of the United Nations 17 Sustainable Development Goals (SDGs) of the 2030 Agenda for Sustainable Development [12]. Some are closely related to the tourism industry, such as SDG 8 (decent work and economic growth), SDG 13 (climate action), SDG 14 (life below water), and SDG 15 (life on land). Overall, SDGs aim to build resilience in contemporary human communities facing disasters, climate change, and human overpopulation. The Economic Commission for Latin America and the Caribbean [12] mentions that the current health crisis can turn towards environmental and social sustainability. For the tourism industry, this can mean modifying the business as usual to prevent environmental damages (e.g., the tourism industry accounts for 5% of global greenhouse gas emissions).

Despite the above, 2021 is also facing new challenges for funding social projects related to marine conservation. Many enterprises' economic support has been put on hold by the health crisis, and research related to COVID-19 has taken funding priority [96, 97]. However, there is the idea of aid packages or stimulus for conservation [96, 98]. In the meantime, we suggest creating bottom-up conservation campaigns by Holbox locals with the support of local governments would play a crucial role in building social and environmental resilience in tourism-related activities restarting under the New Normal.

## Tourists and seafood consumption

Overall, few tourists made decisions based on their potential environmental impact during their vacations (Fig 6E). Tourists that did demonstrate concern for environmental impact did however share a widespread effort of carrying on sustainable food consumption. For the case of seafood, tourists had little information on the sustainability of Holbox's fisheries (Fig 7A). This issue is of significance to Holbox since tourism ignited demand for fish species that lacked commercial value, such as chopas (tripletail, *Lobotes surinamensis*), macabi (bonefish, *Albula vulpes*, near threatened IUCN), tambor (black drum, *Pogonias cromis*), pez loro (parrotfish, *Sparisoma spp.*), and tilapia (*Oreochromis sp.*) (Rubio-Cisneros et al. 2019). Locals prepare "ceviche" (a fish plate) for thousands of tourists that go on whale shark tours with these species. Tourists that are not informed on Holbox's fisheries sustainability matters for local seafood sustainability. Firstly, fishery resources have a role in supporting local biodiversity, and limited information exists on their population dynamics [99]. Secondly, the diversity in Holbox's fishery resources has decreased (common catches include grunts, corvinas, snooks, snappers, and groupers). Thirdly, tourism development has modified some of Holbox's historical fishing grounds [3, 4, 100, 101].

Richter & Klöckner (2017) [102] suggest that using education to link consumers to fisheries exploitation issues can help promote sustainable seafood consumption. The bottom-up conservation campaigns previously mentioned can also be educational resources to inform fishers, tourism stakeholders, and tourists on overfishing and regional fishing bans. Additionally, this could emphasize to consumers that mitigating unsustainable seafood consumption is the responsibility of the whole community of consumers and is possible using efforts taught by these campaigns. This information can further be passed to tourists in other geographic regions under similar circumstances (e.g., ABT). For example, Costa Rica developed a guide for seafood labeling, allowing fishers to know about fish ecology and biology [103]. Fragments of Hope in Belize and the Belize Fisheries Department also created a seafood guide in 2016 to promote sustainable fish consumption and closure seasons [104]. For Mexico, the Marine Stewardship Council certified lobster fishery in the regions of Sian Ka'an and Banco

Chinchorro for many years [105]. This process allowed locals to gain knowledge on the importance of sustainable fisheries. The suspension of this certification was due to the high economic expenses the fishermen had to cope with to get it, however they still maintain a sustainable fishery of this species.

Another option that has high potential to help increase public awareness of the socio-ecological benefits of sustainable seafood consumption is a poster campaign. In the field, we recorded one restaurant with a poster made by CONANP showing the fishing bans. Though information exists, resources are usually scarce for widespread dissemination.

The results from this study help clarify to management authorities the timely need to communicate with restaurant stakeholders to promote sustainable seafood consumption, given that tourists showed a high prevalence of both expecting to eat seafood and actually eating seafood (Fig 7B and 7C), despite having little knowledge of its source (Fig 7D). A challenge for promoting sustainable seafood consumption on Holbox is the existing non-compliance towards fisheries regulations, law enforcement, and illegal fishing [106–108].

The fishing sector in Quintana Roo saw detrimental impacts from the COVID-19 health crisis. Fishing communities faced market closures, problems with product transportation, and a lack of food and health resources [109]. Fishers in the region requested authorities to consider flexibility in fishing closure periods in the face of the health crisis. Fishers also demonstrated concern for widespread illegal fishing. The pause of fishing activities also stopped community surveillance (this had already occurred before the Pandemic) [109]. The government also authorized self-consumption fishing for tourism stakeholders with a Quintana Roo vessel [110]. The closure of tourism activities in Quintana Roo during the initial months of the Pandemic caused more than 20,000 vessels of tourist stakeholders to stay in port [111] (Figs 2 and 3).

## Ecosystem services

Holbox evinces a known phenomenon of how tourism pressures on geographically limited natural resources ignite increasing pollution, freshwater consumption, and habitat degradation. Examples exist worldwide such as in ABT, Panama; Calamianes Islands in the Philippines; Tenerife, Spain; and Barbados in the Lesser Antilles of the West Indies [5, 73, 74, 112].

Results show most tourists that visit Holbox come from Europe and the United States (Fig 1). These places have an early history of nature conservation compared to Mexico. This is exemplified by the 1900's treaties for migratory birds [113]. These countries also have well-established programs for Payments for Ecosystem Services (PES), where citizens participate in preserving landscapes and biodiversity. Our results suggest that the geographic origin of Holbox's tourist population can represent a positive attribute for developing a local PES system.

The current scenario is a challenge given Mexico's top-bottom regulatory scheme, which in many cases, is not entirely successful for environmental protection and the communities' well-being. Mexico has successful PES programs established in early 2000's: The Program of Payment for Hydrological Environmental Services of Forests (PSAH), which consists of direct payments to landowners with forests in good state of conservation and The Program for the Development of Markets for the Ecosystem Services of Carbon Sequestration, the Derivatives of Biodiversity, and to Promote the Introduction and Improvement of Agro-forestry Systems (PSA-CABSA) [114–116].

These PES programs have existed since the early 2000's. They are an excellent example for developing a tailored PES program for Holbox to preserve biodiversity and habitats, and aid in waste disposal. Our results indicate that tourists consider themselves environmentally conscious (Fig 6C), which can facilitate their willingness to pay for conservation. Holbox's locals'

organization for protection is another attribute for promoting local participation in a PES system that can also be supported by local NGO's. This issue presents a timely concern given Holbox's rapidly growing and unorganized urban and tourist development threatening the island's natural capital [3, 4, 101].

Increasing difficulties exist in Mexico to overcome budget cuts in research and environmental protection, challenging the country's conservation commitments [117–119]. Reductions in Mexico's budget for biodiversity conservation translates into less enforcement for tour operators while developing ecotourism activities (for example, whale sharks' tours within the Mexican Caribbean Biosphere Reserve and the Whale Shark Biosphere Reserve) and reduced surveillance to prevent illegal fishing of vulnerable species (see SEMARNAT-NOM059) in MPAs of the Mexican Caribbean [120].

Holbox's local PES scheme could (1) help improve the past and current reality of limited funds to enforce and protect local nature and (2) promote to build up resilient community with compliance for preserving their natural capital and (3) overcome challenging events for future generations facing health crises such as COVID-19 and climate change.

## Similarities of tourists' perspectives between Holbox and ABT

Our results regarding Holbox tourists' perceptions of the environment are similar to those obtained on the ABT (Figs 6 and 7). Similarities most likely occur since both sites face a tourist population with matching geographic origin. Most tourists interviewed in the ABT were European (47%; n = 177) as on Holbox 37% were European and 20% from the United States (n = 225). For details on the ABT results, see [5].

Both sites document a lack of tourists' knowledge of environmental issues of the destination they visit (Figs 6F and 9). These results can help local and international managers address the need for environmental education campaigns. These solutions will be key in both the destination site and in countries in which the main tourist population originates from. These campaigns can help instill "a priori" the tourist population on sound environmental practices when visiting islands and local or regional threats.

## Challenges or limitations of our work

This work offers baseline information on tourists´ perceptions of the environment on an island that in the last two decades transformed into a touristic hotspot in the Mexican Caribbean and faces environmental degradation and social conflict. The results presented here provide insight to policy-makers and tourism managers on the need to build support for programs that transition to a culture of informed islanders and tourists. Building up this new culture by integrating the local community can improve human-nature interactions associated with challenges from unorganized tourism development.

However, a limitation of our research includes the reality which involves the continuing tourism development as is, in a swift phase and the transition to environmentally friendlier tourism practices to be slower, null, or hampered by the lack of funding or vision of local and national authorities to promote it. Our last fieldwork season on Holbox was in November 2021, and tourism activities are now fully active. However, there is a tremendous problem with an increasing amount of trash and its disposal and increasing numbers of people visiting the island (mentioned earlier). Based on our field observations, Holbox´s landscape is not changing for the better.

Furthermore, our results show tourists informed on environmental threats of the sites they visit could help reduce the gap of the missing information seen in our results that acts as a potential barrier to establishing a full understanding of local environmental threats [5] (Fig 9).

Island conservation and environmental education campaigns beyond the local scale exist and can help as Holbox guidelines. These campaigns could promote sustainability knowledge, allowing resilient communities to face health crises such as COVID-19 or environmental disasters. Additional tools, such as social media, could help disseminate the message of, for example, ¨good practices when visiting tropical islands¨ or ¨consuming sustainable seafood when visiting islands¨ via accounts of tourism authorities in the different participating countries. High connectivity and communication between countries became crucial in the COVID-19 health crisis for travel affairs. Increasing communication regarding information about the travel destination can facilitate interest and awareness among tourists to empathize with the environment and human communities they visit.

Despite the above, another limitation of our study is how real it can be to change tourists' willingness to become informed tourists? And how will authorities and society support a change towards investing in forming informed tourists? Our field observations on Holbox on November 2021 showed how most tourists were not complying with COVID-19 regulations. For example, most tourists were not wearing a mask when wearing a mask was mandatory.

## Conclusions

Results on tourists' perspectives reveal that a change between the behaviors present in the "tourist-nature" relationship is crucial for achieving conservation goals related to preserving coastal ecosystem services, which are critical for the tourism industry. In the New Normal, becoming an "informed tourist" is crucial and necessary for traveling. This is essential if citizens from developed countries continue to be increasingly interested in visiting the tropics and consider themselves environmentally conscious, as documented here.

## Acknowledgments

The authors acknowledge student Chapin Dorsett from Whitman College for her help in the field on Holbox Island. The authors acknowledge students Izaizel Libertad Cruz Gómez from Universidad Nacional Autónoma de México, Facultad de Ciencias, and Brenda Correa-Villa from Universidad Autónoma de Nuevo León, Facultad de Ciencias Biológicas for helping Dr. Rubio in data organization.

## Author Contributions

**Conceptualization:** Jorge L. Montero-Muñoz, José I. González-Rojas.

**Formal analysis:** Nadia T. Rubio-Cisneros, Jorge L. Montero-Muñoz.

**Funding acquisition:** Nadia T. Rubio-Cisneros.

**Investigation:** Nadia T. Rubio-Cisneros, Igor I. Rubio-Cisneros.

**Methodology:** Nadia T. Rubio-Cisneros.

**Project administration:** Nadia T. Rubio-Cisneros.

**Resources:** Sara Morales-Ojeda, Mónica Pech, Gabriel Ruiz-Ayma, Marisol Rueda-Flores, Armando Jiménez, Karen Fuentes, José I. González-Rojas.

**Visualization:** Igor I. Rubio-Cisneros, Sara Morales-Ojeda, Armando Jiménez, José I. González-Rojas.

**Writing – original draft:** Nadia T. Rubio-Cisneros, Jorge L. Montero-Muñoz, Igor I. Rubio-Cisneros, Mónica Pech, Rachel Baker, José I. González-Rojas.

**Writing – review & editing:** Nadia T. Rubio-Cisneros, Jorge L. Montero-Muñoz, Igor I. Rubio-Cisneros, Mónica Pech, Gabriel Ruiz-Ayma, Marisol Rueda-Flores, Rachel Baker, José I. González-Rojas.

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
