## [Decision Letter · Decision Letter 0]

20 Oct 2021

PONE-D-21-11653

The need to address knowledge gaps of Human actions on Caribbean touristic islands facing environmental threats: Tourists’ Perceptions of the environment on Holbox Island and archipelago of Bocas del Toro before the COVID-19 health crisis

PLOS ONE

Dear Dr. Nadia Tamata Rubio-Cisneros,

Thank you for submitting your manuscript to PLOS ONE. After careful consideration, we feel that it has merit but does not fully meet PLOS ONE’s publication criteria as it currently stands. Therefore, we invite you to submit a revised version of the manuscript that addresses the points raised during the review process.

We look forward to receiving your revised manuscript.

Kind regards,

Lóránt Dénes Dávid, PhD

Academic Editor

PLOS ONE

Journal Requirements:

5. Please upload a new copy of Figures 2a, 2b, 3, 4, 5, and 6  as the detail is not clear. Please follow the link for more information: " ext-link-type="uri" xlink:type="simple">https://blogs.plos.org/plos/2019/06/looking-good-tips-for-creating-your-plos-figures-graphics/"
" ext-link-type="uri" xlink:type="simple">https://blogs.plos.org/plos/2019/06/looking-good-tips-for-creating-your-plos-figures-graphics/"

6. We note that Figure 1 in your submission contain map images which may be copyrighted. All PLOS content is published under the Creative Commons Attribution License (CC BY 4.0), which means that the manuscript, images, and Supporting Information files will be freely available online, and any third party is permitted to access, download, copy, distribute, and use these materials in any way, even commercially, with proper attribution. For these reasons, we cannot publish previously copyrighted maps or satellite images created using proprietary data, such as Google software (Google Maps, Street View, and Earth). For more information, see our copyright guidelines: http://journals.plos.org/plosone/s/licenses-and-copyright.

Reviewers' comments:

Reviewer's Responses to Questions

**Comments to the Author**

1. Is the manuscript technically sound, and do the data support the conclusions?

Reviewer #1: Yes

Reviewer #2: Yes

2. Has the statistical analysis been performed appropriately and rigorously? 

Reviewer #1: Yes

Reviewer #2: Yes

3. Have the authors made all data underlying the findings in their manuscript fully available?

Reviewer #1: Yes

Reviewer #2: Yes

4. Is the manuscript presented in an intelligible fashion and written in standard English?

Reviewer #1: Yes

Reviewer #2: Yes

5. Review Comments to the Author

Reviewer #1: The paper is about a very current topic. The research aims and the reletad metholodogy are clear and proper. The reserach is based on a detailed and relevant literature review. THa results of the paper have high novelty. I recommend the paper for acceptance in current form.

Reviewer #2: The paper focuses on a contemporay and actual topic. The title is too long, should be shortened or compressed somehow. The abstract is well written and the keywords are well selected.

The literature review is too short, should be extended.

The results are clear, well supported by the methodological toolset. LImitations of the study is not described.

6. PLOS authors have the option to publish the peer review history of their article (what does this mean?). If published, this will include your full peer review and any attached files.

Reviewer #1: No

Reviewer #2: No

---

## [Author Response · Author response to Decision Letter 0]

4 Dec 2021

Nadia T. Rubio-Cisneros, Ph.D.

Director and Founder

Mar Sustentable Ciencia y Conservación, A.C.

https://marsustentable.org

Academic Affiliations

Universidad Autónoma de Nuevo León

Laboratorio de Biología de la Conservación.

Associated Researcher, Center for 

Marine Biodiversity and Conservation, 

SIO, UCSD

April 8, 2021

PLOS ONE

Editorial Committee

Dear Editorial Committee, it is a pleasure to meet you. I am submitting the reviewers comments for our paper entitled: ¨The need to address knowledge gaps of Human actions on Caribbean touristic islands facing environmental threats: Tourists' Perceptions of the environment on Holbox Island and archipelago of Bocas del Toro before the COVID-19 health crisis, which is in review for publication in PLOS ONE. 

Response to Reviewers

Reviewer #1: 

The paper is about a very current topic. The research aims and the related methodology are clear and proper. The research is based on a detailed and relevant literature review. The results of the paper have high novelty. I recommend the paper for acceptance in its current form. 

Reviewer #2: 

• The paper focuses on a contemporary and actual topic. 

• The title is too long, should be shortened or compressed somehow. 

The title was shortened to 

Tourists perceptions of Caribbean islands facing environmental threats before the COVID-19 health crisis: Holbox Island and Archipelago of Bocas del Toro 

• The abstract is well written, and the keywords are well selected.

The literature review is too short, should be extended.

• The authors added several current references that supported our work ideas. Below I mention them:

Abbas, J., Mubeen, R., Iorember, P. T., Raza, S., Mamirkulova, G. (November 01, 2021). Exploring the impact of COVID-19 on tourism: transformational potential and implications for a sustainable recovery of the travel and leisure industry. Current Research in Behavioral Sciences, 2.

Balzan, Mario Potschin-Young, Marion Haines-Young, Roy. (2018). Island Ecosystem Services: insights from a literature review on case-study island ecosystem services and future prospects. International Journal of Biodiversity Science, Ecosystem Services Management. 14. 71-90. 10.1080/21513732.2018.1439103.

Brinklow L, Ellsmoor J, Randall J, Rouby M, Sajeva G, Shetye A. Sindico F. COVID 19 Island Insights Series Final Report. University of Strathcyce. Centre for Environmental Law and Governance. 2021; 3-160.

Hakim, L. (2020). COVID-19, tourism, and small islands in Indonesia: Protecting fragile communities in the global Coronavirus pandemic. Journal of Marine and Island Cultures, v9n1, 130-141. https://doi.org/10.21463/jmic.2020.09.1.08

https://webunwto.s3.eu-west-1.amazonaws.com/s3fs-public/2020-08/SG-Policy-Brief-on-COVID-and-Tourism.pdf

https://www.palcoquintanarroense.com.mx/noticias-de-quintana-roo/habitantes-de-holbox-abandonan-pesca-para-dedicarse-de-lleno-a-actividad-turistica/

Jouault, S., Rivera-Nuñez, T., García de Fuentes, A., Xool Koh, M., Montañez Giustinianovic, A. (2021). Respuestas, resistencias y oportunidades del turismo comunitario en la península de Yucatán frente al COVID-19 y las crisis recurrentes. Investigaciones Geográficas, (104). https://doi.org/10.14350/rig.60240

Lopez-Ercilla, Ines, Maria Jose Espinosa-Romero, Francisco J. Fernandez Rivera-Melo, Stuart Fulton, Rebeca Fernández, Jorge Torre, Araceli Acevedo-Rosas, Arturo J. Hernández-Velasco, and Imelda Amador. 2021. "The voice of Mexican small-scale fishers in times of COVID-19: Impacts, responses, and digital divide". Marine Policy. 131: 104606.

Mulder, N (coord.), “The impact of the COVID-19 pandemic on the tourism sector in Latin America and the Caribbean, and options for a sustainable and resilient recovery”, International Trade series, No. 157 (LC/TS.2020/147), Santiago, Economic Commission for Latin America and the Caribbean (ECLAC), 2020.

Palco Noticias. 2021. Habitantes de Holbox abandonan pesca para dedicarse de lleno a actividad turística. Palcoquintanarroense.com.mx. November 3, 2021.

Sheller, M. (2021) Reconstructing tourism in the Caribbean: connecting pandemic recovery, climate resilience and sustainable tourism through mobility justice, Journal of Sustainable Tourism, 29:9, 1436-1449, DOI: 10.1080/09669582.2020.1791141

United Nations. 2020. COVID-19 and Transforming Tourism. United Nations.

• The results are clear, well supported by the methodological toolset. 

Limitations of the study is not described. 

• The authors modified the previous conclusion section to add some examples of limitations that our study can face.

Other modifications made to the manuscript are mentioned below.

• The figures were updated in the required format using the PASE program.

• All the figures developed for this study were from the original idea of first author Nadia Rubio and created with our data sources.

• The reference list was updated.

Sincerely 

Nadia Rubio-Cisneros, Ph.D.

---

## [Editor Report · Decision Letter 1]

16 Dec 2021

Tourists perceptions of Caribbean islands facing environmental threats before the COVID-19 health crisis: Holbox Island and Archipelago of Bocas del Toro

PONE-D-21-11653R1

Dear Colleague,

We’re pleased to inform you that your manuscript has been judged scientifically suitable for publication and will be formally accepted for publication once it meets all outstanding technical requirements.

Kind regards,

Lóránt Dénes Dávid, PhD

Academic Editor

PLOS ONE

Additional Editor Comments (optional):

Accepted.
---

## [Editor Report · Acceptance letter]

28 Feb 2022

PONE-D-21-11653R1 

Tourists perceptions of Caribbean islands facing environmental threats before the COVID-19 health crisis: Holbox Island and Archipelago of Bocas del Toro  

Dear Dr. Rubio-Cisneros:

I'm pleased to inform you that your manuscript has been deemed suitable for publication in PLOS ONE. Congratulations! Your manuscript is now with our production department. 

Kind regards, 

on behalf of

Dr. Lóránt Dénes Dávid 

Academic Editor

PLOS ONE